# Filamentogenesis and Filamentolysis of a Light Filament: Dynamic Processes in the Near-Surface Ocean Under Tidal Forcing

Michelle Albinus<sup>1</sup>, Thomas H. Badewien<sup>1</sup>, Lisa Gassen<sup>2</sup>, Oliver Wurl<sup>2</sup> and Jens Meyerjürgens<sup>1</sup>

<sup>1</sup>Carl von Ossietzky Universität, Institute for Chemistry and Biology of the Marine Environment, Marine Sensorsystems, Oldenburg, Germany.

<sup>2</sup>Carl von Ossietzky Universität, Institute for Chemistry and Biology of the Marine Environment, Processing and Sensing of Marine Interfaces, Oldenburg, Germany.

Correspondence to: Michelle Albinus (michelle.albinus@uol.de), Jens Meyerjürgens (jens.meyerjuergens@uol.de)

Abstract. This study investigates the dynamics and evolution of a light filament embedded within a tidal mixing front, focusing on its spatial and temporal evolution in the near-surface layer (0.1 - 10 m) of the water column. A high-resolution, multi-sensor synoptic dataset, consisting of surface drifters, a drifting sensor chain, and an autonomous surface vehicle equipped with an Acoustic Doppler Current Profiler, temperature, and conductivity sensors, was used to observe patterns of divergence, vorticity, and vertical velocities, avoiding high temporospatial aliasing. The measurements resolved three phases of the filament occurring on length scales of O(0.1-2 km) and time scales of minutes to one hour: I) establishment of the filament in the overlying first meter and filamentolysis 




The southern North Sea provides an ideal setting to study these processes. There, tides, wind forcing, and freshwater inputs create a highly energetic and deformable frontal environment. Tidal mixing fronts (TMFs) form at the interface between vertically stratified and well-mixed waters in summer, when surface heat flux steepens lateral temperature gradients (Van Heijst, 1986). TMFs occur on continental shelves worldwide (Timko et al., 2019) and mediate exchanges of nutrients, particles, and energy between ocean layers (Thomas et al., 2008; Sun and Cho, 2010). In the North Sea, TMFs vary seasonally under semidiurnal tides, fortnightly spring—neap cycles, bottom topography, and intermittent winds (Simpson et al., 1990; Holt and Umlauf, 2008; Zhao et al., 2019). Freshwater from rivers such as the Elbe can lower coastal salinity by up to 10 g·kg<sup>-1</sup>, steepening density gradients and intensifying fronts (Simpson et al., 1990; Ricker et al., 2021; Goßmann et al., 2025). Shallow bathymetry amplifies these effects, making TMFs highly sensitive to mesoscale and submesoscale strain, a key prerequisite for frontogenesis (Garcia-Jove et al., 2022).

During frontogenesis, ASC drives upward motion on the light side and downward motion on the dense side, flattening isopycnals and enhancing stratification, while reversed ASC leads to frontolysis (McWilliams, 2009; Chrysagi et al., 2021; Garcia-Jove et al., 2022). Similar dynamics govern filaments, referred to as filamentogenesis and filamentolysis (e.g., McWilliams 2009, 2015; Garcia-Jove et al., 2022; Jakes et al., 2023). Dense filaments are reinforced by ASC-driven convergence, while light filaments experience reversed circulation, promoting rapid decay. Further, dense filaments can extend down several hundred meters (e.g., Garcia-Jove et al., 2022), while light filaments are rather pronounced at the ocean surface (Thomas et al., 2008). Capturing these dynamics requires high-resolution measurements and calculation of Differential Kinematic Properties (DKP), such as divergence, vorticity, and horizontal velocity gradients, from water-column instabilities or parcel deformation (e.g., Vélez-Belchí and Tintoré, 2001; Molinari and Kirwan, 1975; Berta et al., 2016; Huntley et al., 2022). Surface drifters, HF radar, and ADCPs have been widely used (e.g., Hill et al., 1993; Shcherbina et al., 2013; Berta et al., 2016, Archer et al., 2020; Esposito et al., 2023), yet most studies lack data from the uppermost meter of the ocean, precisely where frontal and light filamentary processes peak (Thomas et al., 2008; Chrysagi et al., 2021; Gassen et al., 2023). Submesoscale features (O[0.1-10 km]) are critical for the ocean's energy cascade, both distributing energy forward and feeding larger scales (Schubert et al., 2020; Zhang et al., 2023). In shallow seas like the North Sea, these processes are assumed to be strongly imprinted at the surface, yet high-resolution observations remain scarce and temporally aliased (Johnson et al., 2024).

Building on this gap, we suggest that tidal oscillations in the southern North Sea, by alternately shearing, stretching, and compressing the density field, may occasionally generate the conditions necessary for light filament formation. While tidal reversals may favour the export of dense filaments offshore, phases where tidal flow and wind-driven advection align may instead pull buoyant surface water into denser surroundings. In these situations, transient anticyclonic strain zones can develop along the front, momentarily reversing the usual dense-offshore export and enabling warm-side water to intrude into



the cold side. Although such light filament events are rare and short-lived, the unique tidal environment of the southern North Sea offers occasional opportunities to capture them. To test this hypothesis, we conducted high-resolution ADCP and CTD measurements at O[0.1-2 km] scales using an autonomous surface vehicle (ASV) at a TMF in the southern North Sea. Trajectories from 14 surface drifters and a drifting sensor chain complemented these observations, resolving the evolution of a light filament with a focus on the near-surface layer (NSL; 0.1-10 m). This dataset allows us to characterize the mechanisms of rapid filament development, surface expression, and decay, providing new insight into these elusive but dynamically important features.

#### 2 Materials and Methods

## 2.1 Data acquisition

During cruise HE626 on RV Heincke, we collected a multi-platform dataset combining ship-based, autonomous, and Lagrangian observations. Central to the study were measurements by the ASV Halobates (Wurl et al., 2024), which provided current velocities in the NSL as well as high-resolution temperature and salinity profiles. Additional underway observations included wind data and continuous NSL hydrographic measurements (temperature and conductivity) from a 4H Jena Engineering Pocketbox Ferrybox flow-through system. Stationary measurements consisted of ship-based Seabird SBE911+ CTD casts and deployments of Lagrangian platforms, including 14 near-surface drifters (Meyerjürgens et al., 2019) and a drifting sensor chain. The focus of this study is on observations from 5 August 2023, northwest of Helgoland (Fig. 1a), targeting submesoscale frontal processes under strong semidiurnal tidal influence. The region was characterized by TMF structures on scales of O(0.1-10 km), visible in sea surface salinity (SSS). The more saline water mass originated from the open North Sea, while the fresher mass consisted of a mixture of North Sea water and river runoff from the German Bight (Burchard and Badewien, 2015). Semidiurnal tides imported saline offshore water, creating horizontal gradients of up to  $\pm 1.5~{\rm g\cdot kg^{-1}}$  (Fig. 1a). The frontal system exhibited submesoscale meanders (Fig. 1a) and developed a light filament on horizontal scales of 0.1-2 km (Fig. 1c).

Front detection was guided by SSS fields from the CMEMS Atlantic–European Northwest Shelf Ocean Physics Analysis and Forecast model (CMEMS, 2024), which has a horizontal resolution of 0.027° × 0.027° and hourly output. For the target region, model SSS correlated reasonably with in situ Ferrybox salinity (R = 0.71; Fig. B1). On the evening of 4 August, the vessel carried out a zig-zag survey (~7 kn) to validate the model fields and identify the strongest gradient, which reached ΔSSS ≈ 1.2 g·kg<sup>-1</sup>·km<sup>-1</sup>. In situ salinity was obtained from the Ferrybox, pumping seawater from 3 m depth (~6 L·min<sup>-1</sup>) with conductivity measurements accurate to ±0.3 mS·cm<sup>-1</sup>. Once the gradient was confirmed, the vessel remained on station to deploy the Lagrangian instruments. A sensor chain equipped with six Sea&Sun CTM 48M CTDs mounted between 0.2-10 m depth, transmitting position data every 2 min (Garmin inReach Messenger GPS), and 14 surface drifters transmitting


positions every 5 min (SPOT Trace® GPS). The drifters were deployed by zodiac in the form of 12 triangular arrays (Fig. 1b) and tracked over 25 h. Their trajectories showed a north-westward drift under prevailing southeasterly winds (Fig. C12), covering ~21 km, while their rotary motion reflected tidal currents (Meyerjürgens et al., 2020). During ebb tide, drifters were displaced westward at speeds up to 1 m·s<sup>-1</sup>, an effect intensified by wind forcing between 14:00-21:00 UTC when wind speeds increased from 6 to 14 m·s<sup>-1</sup>. The sensor chain largely followed the drifter displacement.

Figure 1. a) The study site is located in the southern North Sea, northeast of Helgoland, highlighted with a red box. On August 5th, 2023, the area exhibits submesoscale structures with SSS gradients of  $\pm$  1.5 g  $\cdot$  kg<sup>-1</sup>  $\cdot$  km<sup>-1</sup>. b) Measurements within the red box include a drifting sensor chain (black dot-dashed line), 14 Lagrangian surface drifter deployments (initial positions as black squares) and the ASV (pink area). Drifter trajectories are colored by current velocity magnitude. c) Density (black line), salinity (blue line) and temperature (red line) of the filament sampled by the ASV.



After the Lagrangian deployments, the ASV HALOBATES (Wurl et al., 2024) was launched within the frontal zone to observe the propagation of the light filament at high temporal resolution. The vehicle executed a repeating rectangular track (~2 km east-west, 0.08 km north-south), with only the east-west transects analyzed (Fig. 1c). Seven transects were completed between 06:30 and 12:45 UTC at ~2 kn. The ASV carried a Teledyne RD Instruments RiverRay 600 kHz ADCP, mounted at 0.275 m immersion depth. With a 1 m vertical bin size, velocity profiles were obtained from 1-10 m depth (excluding near-bottom bins). Thus, ADCP data began at 1 m and extended through the NSL, but did not cover the overlying surface layer. In contrast, the ASV was equipped with seven Idronaut OS310 CTDs mounted at 80-100 μm (denoted the skin layer; Shinki et al., 2012), 30 cm, 40 cm, 50 cm, 60 cm, 85 cm, and 100 cm. Those resolved temperature and salinity changes within the skin layer down to one meter of the water column, allowing the evolution of the filament surface expression to be captured in detail. HALOBATES is equipped with rotating glass plates that pick up the skin layer by surface tension followed by scrapes transferring the water into the tube system being connected to the CTDs. Because of this setup, the CTDs and ADCP provided complementary rather than overlapping coverage: CTDs focused on the upper meter, while the ADCP recorded current velocities in the deeper part of the NSL.

25 For analysis, ADCP data were filtered to remove outliers (>3σ from the mean), truncated below 10 m depth, and smoothed using a 60 s moving average. Drifter positions were post-processed following Deyle et al. (2024), yielding interpolated trajectories with a uniform 5 min resolution, including bridging of GPS gaps up to 1 h.

# 2.2 Surface divergence from drifters

A variety of studies have estimated divergence by tracking the deformation of drifter clusters (Molinari and Kirwan, 1975; Kawai, 1985; Berta et al., 2016; Tarry et al., 2021; Huntley et al., 2022; Esposito et al., 2023). Molinari and Kirwan (1975) introduced two robust approaches: (a) the method of least squares, which infers divergence from velocity gradients fitted around the cluster centroid, and (b) the area-based method, which calculates divergence from the rate of change in the polygonal area spanned by a drifter group. While the least-squares method scales well to larger clusters and offers higher statistical confidence (Molinari and Kirwan, 1975; Tarry et al., 2022), it requires more drifters and is sensitive to tidal modulation of trajectories, which can obscure frontal signatures. The area-based method is less demanding, relying on only three drifters, and is therefore better suited for frontal environments where deformation rates are high. In this study, we applied the latter, expressed as

$$\delta = \frac{1}{A} \frac{\partial A}{\partial t} \tag{1}$$

where A is the area of a drifter triplet and  $\partial A/\partial t$  its temporal change.

150

155

To maximize the number of usable triplets, drifters were deployed on 5 August 2023 as vertices of 12 potential triangles with edge lengths of 0.2-0.5 km along a ~4 km transect (Fig. 1b). Only complete triplets were retained, excluding 21.8% of deployments with <3 drifters. Because high-gradient fronts coincide with elevated strain rates (Mahadevan and Tandon, 2006), clusters in such areas were particularly prone to rapid elongation and biased divergence estimates. To address this, Berta et al. (2016, 2020) and Huntley et al. (2022) proposed quality metrics that characterize triangle geometry. One such measure is

$$\Lambda = \frac{12\sqrt{3}A}{P^2} \tag{2}$$

where P is the perimeter.  $\Lambda = 1$  corresponds to equilateral triangles, while  $\Lambda = 0.2$  indicates near-collinear configurations. Huntley et al. (2022) demonstrated that  $\Lambda$  provides slightly more reliable results than internal-angle metrics  $(\theta, \gamma)$ , with ~4 % better accuracy. Given the limited sample size (~20 triplets per time step), this improvement was significant. Applying a conservative filter of  $0.2 \le \Lambda \le 1$  retained 36.4 % of the available clusters, leaving 3559 valid triplets for divergence estimation. Overlaps in drifter trajectories in highly dynamic regimes (e.g., tidal transitions or wind-driven surges; Fig. 2b) further reduced the number of suitable triplets (Berta et al., 2016; Villa Castrillón et al., 2024).

Divergence was then calculated for each triplet centroid using equation (1). Triplets were time-binned according to tidal phase, since tidal reversals prevented uniform resampling. Five intervals were identified (t<sub>1</sub>-t<sub>5</sub>: 1.5, 5, 7.5, 5.5, 5.5 h). Delaunay triangulation at the start of each interval increased usable triplets and avoided configurations already deformed. Divergence estimates were subsequently gridded onto a 1 × 1 km grid, averaging values within each spatial bin and time interval. This yielded 2179 triplets suitable for analysis. Finally, divergence values were normalized by the Coriolis frequency  $f = 1.18 \times 10^{-4} \, \text{s}^{-1}$  at a reference latitude of 54.25 °N.

## 2.3 Water column properties and instabilities from drifting sensor chain and ASV CTDs

Since the sensor chain trajectory broadly followed the drifters, it provided an additional basis to investigate water column structure and instabilities at the front. To compare with the drifter-derived divergence field, divergence was also estimated from the chain trajectory. Coordinates were first smoothed with a 10 min moving average to reduce noise, and divergence δ was then calculated as

$$\delta = \frac{\partial u}{\partial x} + \frac{\partial v}{\partial y} \tag{3}$$



where u and v are the cross- and along-front velocity components, obtained by rotating earth-referenced velocities into the chain's azimuthal frame. As before,  $\delta$  was normalized by the Coriolis frequency f.

To characterize the water masses shaping the front, temperature and conductivity from the chain sensors were converted to conservative temperature, absolute salinity, and in-situ density at depths z = [0.2, 1.2, 6.2, 8.4, 10.6] m. The same procedure was applied to the ASV CTD data at depths z = [0.0001, 30, 40, 50, 60, 85, 100] cm. These profiles were then used to derive water column instability proxies: the Brunt-Väisälä frequency

$$N^2 = \frac{\partial b}{\partial z},\tag{4}$$

and the lateral buoyancy gradient

$$M^2 = \frac{\partial b}{\partial x'} \tag{5}$$

with buoyancy defined as  $b = g(1 - \rho / \rho_0)$ , where  $\rho$  is potential density and  $\rho_0 = 1022.3$  kg m<sup>-3</sup> is the surface reference density. By examining the evolution of isopycnal slopes, periods of restratification and mixing could be identified.

#### 2.4 Divergence from ASV-mounted ADCP

To investigate frontal dynamics at the lower end of the submesoscale and at high temporal resolution, divergence was also calculated from the ADCP mounted on the ASV. Following Rudnick (2001), the assumption of a frontal jet aligned with the front allows equation (3) to be simplified to

$$\delta \approx \frac{\partial u}{\partial x},\tag{6}$$

a formulation commonly known as the "one-ship method" (Shcherbina et al., 2013). This approach has been widely applied to ADCP data for single-depth divergence estimates (e.g., Drinkwater and Loder, 2001; Archer et al., 2020; Esposito et al., 2023).

From the ASV-mounted ADCP, the along-track velocity component was extracted and organized into transects (Fig. 1c).

Data were averaged onto a grid with 10 m horizontal spacing, 2 min temporal resolution, and 1 m vertical intervals before applying equation (6), and the resulting divergence estimates were normalized by the Coriolis frequency f. To capture the





temporal evolution at TMF, full-depth transects were analyzed at three times during the morning sequence (07:23, 07:40, 08:30). In addition to horizontal divergence, the vertical component of relative vorticity,

$$\zeta = \frac{\partial v}{\partial x} - \frac{\partial u}{\partial y},\tag{7}$$

was derived using the same method, together with the along-front velocity v, to characterize the jet structure. Surface density transects from the ASV CTD at z=1 m were analyzed for the same times, alongside co-located depth sections of  $\delta$ ,  $\zeta$ , and v. Vertical velocity fields were overlaid to highlight frontogenesis and frontolysis processes (see Section 2.5).

## 2.5 Vertical velocity from drifter and ADCP divergence estimates

Since horizontal divergence was used to diagnose vertical motion, vertical velocities in the water column were derived from the continuity equation,

$$\frac{\partial u}{\partial x} + \frac{\partial v}{\partial y} + \frac{\partial w}{\partial z} = 0$$

which can be rearranged to

$$\Delta w = -\int_{-z}^{z_0} \left(\frac{\partial u}{\partial x} + \frac{\partial v}{\partial y}\right) dz,\tag{8}$$

where  $\Delta w$  is the vertical velocity difference between the surface ( $z_0$ ) and depth z. For drifters, the target depth was set to z=-0.5 m, corresponding to the drifter's immersion depth (Meyerjürgens et al., 2019). At the surface, previous studies (Tarry et al., 2021; Rypina et al., 2021; Tarry et al., 2022; Esposito et al., 2023) assumed  $w(z_0=0)=0$  m d<sup>-1</sup>, as vertical motion at the surface is typically one to two orders of magnitude smaller than observed subsurface velocities. Rypina et al. (2021) argued that sea surface displacement is largely governed by gravity and tidal waves, yielding negligible vertical motion in the Mediterranean.In contrast, the North Sea exhibits much larger tidal amplitudes. Around Helgoland, the M2 tide causes vertical displacements of ~1.5 m (Stanev et al., 2014), making the tidal-induced vertical velocity at the surface comparable to observed w. Thus, the assumption  $w(z_0=0)=0$  m d<sup>-1</sup> was not valid here. Instead, sea surface height (SSH) with a routmean-square difference (RMSD) of ~ 5 - 15 cm from CMEMS model data (2024) was used to estimate tidal vertical velocities at  $z_0$ , applied as hourly means. For ADCP data, vertical velocities were calculated across the full depth range using equation (8). In this case, the one-ship method (equation 6) was substituted for the full divergence term, yielding:

$$w(z_i) = w(z_{i-1}) + \left(\frac{du}{dx}\right)\Delta z,\tag{9}$$

where du/dx is the along-track divergence and  $\Delta z$  the ADCP bin size. A detailed derivation of equation (9) is provided in Appendix A. Since ADCP measurements begin at 1 m depth, tidal integration at the surface was unnecessary; the first estimate of w could be calculated directly at 2 m.

# 3 Results






## 3.1 Water column characteristics and instabilities from drifting sensor chain

During the high tide period between 09:00 and 12:00, the sensor chain crossed the light filament, which was observed from the surface down to ~4–5 m and exhibited a density about 0.4 g kg<sup>-1</sup> lower than the surrounding TMF. It was embedded at the lower boundary of the TMF salinity signal (~31.5 g kg<sup>-1</sup>) and at the highest boundary of the TMF temperature signal (~19.0 °C), confirming its formation on the light side of the front, as hypothesized. The filament persisted for approximately 3 h, showing a two-layer stratification from 09:00 to 12:00, with fresh (warm) water overlying more saline (cold) water. During the first two hours, the upper, lighter layer extended down to ~4 m and retreated to ~2 m in the final hour. This retreat was driven by a decrease in the salinity gradient while the temperature gradient increased (Fig. 2b,c). Around 17:00, the sensor chain crossed to the dense side of the TMF, where the filament was no longer apparent in salinity, temperature, or density.

The filament also influenced water column stability, evident in the Brunt-Väisälä frequency ( $N^2$ ) and lateral buoyancy gradient ( $M^2$ ). The light filament emerged from the surrounding TMF as a region of strongly positive  $N^2$  (Fig. 2d), indicating stable stratification. Below 7 m, negative  $N^2$  values suggested local buoyancy production, likely caused by vertical shear acting on horizontal density gradients near the bottom, possibly due to tidal interactions with topography. This feature correlated with the bifurcation of isopycnals ("forking"), where dense water was forced over lighter water, a property appearing along the entire TMF time series rather than being unique to the filament. Considering the lateral buoyancy gradient (Fig. 2e),  $M^2$  was positive (stable) when the filament was at ~4 m depth, but reversed to negative values during its retreat, indicating buoyancy production and the upward displacement of the filament, as will be further discussed in a later section.



Figure 2. Time series of a) absolute salinity, b) conservative temperature, c) potential density, d) Brunt-Väissäla frequency  $N^2$  and e) lateral buoyancy gradient  $M^2$  obtained from the drifting sensor chain during August 5th and 6th. Vertical black dashed lines mark low water (LW) and high water (HW).  $N^2$  and  $M^2$  are displayed with overlain isopycnals.

## 3.2 Coupling of filamentary and tide-induced vertical motions

Figure 3 shows drifter triplet centroid trajectories over SSS isohalines (CMEMS, 2024, panel a), along with space- and time-binned near-surface horizontal divergence normalized by f (Fig. 3b) and vertical velocity w (Fig. 3c). As the drifters crossed TMF isohalines, these data reveal across-front divergence and corresponding areas of up- and downwelling. Both SSS and drifter trajectories indicate that the TMF meandered on the mesoscale along the tidal wave direction, while a sudden southward diffraction of the northern trajectories around 8.05 °E longitude suggests a submesoscale light filament.

Following the main high tide current eastward (Fig. 3a), most drifters initially diverged with  $\delta \approx 2f$  due to accelerating tidal flow (Fig. 3b, Fig. 1b). Shortly before 10:00, the northern cluster entered a convergence zone ( $\sim$  -2f), then peaked at 5f after  $\sim$ 30 min, and dropped to -5f around 11:00. Although  $\delta \approx \pm$  5f is high relative to previous



studies (Berta et al., 2016; Tarry et al., 2021, 2022), strong triplet deformation due to the tides likely caused overestimation (see section 2.2). Divergence drove light filament water upward away from the axis, while dense water sank during convergence (Fig. 3c), with vertical velocities ranging from  $\sim -30 \, \text{m d}^{-1}$  to  $50 \, \text{m d}^{-1}$ . Despite potential overestimation due to error propagation from the divergence estimation, the vertical velocity pattern aligns with the drifter centroid trajectories, including the southward diffraction at 8.05 °E longitude. This behavior indicates the northern cluster sampled a light filament, consistent with sensor chain observations (Fig. 2). The steepening of isopycnals (Fig. 2c) matches downwelling, while the sensor chain's decreasing vertical filament extent coincides with drifter-indicated upwelling.

Figure 3. a) Trajectories of the centroids from drifter triplets over latitude and longitude colored according to time on August 5th.

The map background displays the corresponding isohales in  $g \cdot kg^{-1}$  (CMEMS, 2024; gray solid lines) as mean over the displayed drifter period. b) Normalized near-surface divergence and c) vertical velocity in  $m \cdot d^{-1}$  for the time sections of drifter triplets in


a) colored according to northern (magenta) and southern (grey) drifter triplet clusters. Black outlines highlight the sampling of the filament.

The southern cluster showed weaker divergence, generally fluctuating between ± 2*f*. At the filament location, these centroids did not display the diffraction seen in the northern cluster and remained on the 29.8 g kg<sup>-1</sup> isohaline during peak tidal flow, whereas the northern cluster crossed diapycnally by ~ -0.5 g kg<sup>-1</sup>. Outside the filament, *w* exhibited an overall up- to downwelling trend for both clusters, reflecting decreasing tidal velocities (Fig. 3b). In regions of filament-induced vertical motion, tides either mediated or amplified *w*. For example, during the first convergence zone, filamentary downwelling was partly offset by tide-induced upwelling, while in the second convergence period, weaker tidal flow enhanced downwelling. These results show that the light filament's evolution was embedded within and dynamically coupled to tidal currents.

## 3.3 Filament structure and evolution of isopycnals in the overlying first meter

During the ebb period, three phases were recognized in the evolution of the light filament in the overlying first meter of the NSL (Fig. 4). Panels a, d, g show phase I, where isopycnals were vertically aligned and density varied by about  $0.4 \text{ kg m}^{-3}$  over less than 0.5 km. Although data are only available for x 

Figure 4. CTD data from the ASV displayed as contour section over depth across the filament at t1 = 07:23 am, t2 = 07:40 am and t3 = 08:30 am, each for of a-c) absolute salinity, d-f) conservative temperature and g-i) potential density in the upper NSL.

In phase II (t2; Fig. 4b, e, h), isopycnals above 0.6 m began to flatten away from the filament axis, and the light core of the filament shifted into the upper layer above 0.6 m, with a slight intensification of about 0.1 kg m<sup>-3</sup> (Fig. 4h). Below 0.6 m, no flattening occurred, but the temperature gradient decreased, with a minor reduction in salinity (Fig. 4b,e). This led to compression of the filament core, reducing its horizontal extent in this layer to roughly half of its initial length. Phase III (t3; Fig. 4c, f, i) occurred less than one hour later. Isopycnals were now nearly horizontal, and the light filament core was distributed homogeneously in the upper 0.6 m (Fig. 4i). Salinity followed the density pattern, while temperature became horizontally stratified with a gradient of ~0.2 °C.







Throughout the observation period, the filament core salinity remained at 30.6 g kg<sup>-1</sup> (Fig. 4a-c), while the surrounding water mass decreased by ~0.4 g kg<sup>-1</sup>. Temperature within the filament increased by ~0.3 °C, compared to ~0.1 °C in the surrounding water (Fig. 4d-f). Salinity primarily controlled density, which decreased by ~0.1 kg m<sup>-3</sup> within the filament and ~0.3 kg m<sup>-3</sup> in the surrounding water (Fig. 4g-i). The observed evolution of the light filament exhibits characteristics reminiscent of filamentogenesis, including upward displacement of the light core and lateral compression. Unlike classical cases documented in the literature (e.g.; McWilliams et al., 2015; Chrysagi et al., 2021; Jakes et al., 2023), these dynamics are driven primarily by salinity gradients rather than temperature gradients, reflecting the strong influence of freshwater input and tidal modulation in the southern North Sea. While ASC patterns are not yet analyzed, these observations suggest the filament is undergoing a process analogous to filamentogenesis, to be further confirmed with ADCP-derived velocity fields in the following section. Notably, the upward displacement of the filament core is consistently observed across all datasets: the sensor chain covering the NSL below 1 m, the drifter trajectories, and the ASV CTD measurements in the upper meter, reinforcing the robustness of this evolution.

## 3.4 ASC patterns and evolution of divergence, vorticity and vertical velocity in the NSL

During phase I of the filament evolution in the NSL, divergence occurred around the Coriolis frequency f, with a significant convergence area between x = 0.5 km and x = 1.5 km (Fig. 5d). Over depth, the convergence core extended from 4 m at  $x \approx 1$  km down to 8 m at x = 1.5 km. Dense water sank along the TMF density gradient (Fig. 5a) at ~15 m d<sup>-1</sup> (Fig. 5j), accumulating beneath lighter water (Fig. D1). This downwelling also dragged light water downward at about half the magnitude of w. Below the convergence zone, divergence developed alongside anticyclonic vorticity for x > 1 km and cyclonic vorticity for x < 1 km, both within the Rossby radius (Fig. 5g; Thomas et al., 2008), producing uplift of the filament flanks at ~12 m d<sup>-1</sup>. CTD profiles (Fig. D1) confirmed dense bottom water being advected upward into shallower layers by divergence flow. These ASC signals correspond to a reversed pattern of light filamentogenesis, i.e. filamentolysis, characterized by downwelling in the core and uplift on the flanks.

In phase II, the ASC signatures shifted. As divergence increased, the convergence core contracted vertically by  $\sim$ 2-3 m and horizontally by  $\sim$ 0.25 km within 17 minutes (Fig. 5e). The steep density gradients relaxed through mixing, reducing downward motion (Fig. 5k), while CTD profiles showed flattening isopycnals at z=100 cm. Anticyclonic divergence expanded toward the surface on the right boundary (Fig. 5e,h), driving upwelling up to 20 m d<sup>-1</sup> (Fig. 5k) and exporting dense water laterally away from the TMF axis. On the left side, strong divergence and anticyclonic vorticity brought dense water upward into the convergence zone (Fig. 5h). Overall, phase II represented a transitional regime: the strong downwelling of phase I largely disappeared, but the complete boundary downwelling characteristic of filamentogenesis had not yet emerged. Instead, ASC signatures combined remnants of filamentolysis with early signs of filamentogenesis, suggesting that the main circulation had already shifted upward into the overlying first meter.

Figure 5. a-c) Density profile over distance from ASV CTD data at z=100 cm for t1 = 07:23 am, t2 = 07:40 am and t3 = 08:30 am on August 5th in the NSL. Underlying panels show corresponding sections over depth of normalized d-f)  $\delta$ , g-i)  $\zeta$  and j-l) along-front velocity with overlying w (arrows) referenced to 20 m d<sup>-1</sup>





By phase III, the ASC patterns below 1 m revealed only the tails of the process. The density at z=100 cm dropped by ~0.3 kg m<sup>-3</sup> on the left side (Fig. 5c), consistent with vertical CTD profiles. During the final 40 minutes, the anticyclonic patch doubled in magnitude and extended through the full water column (Fig. 5i), while a cyclonic patch developed at x 





relative to the surrounding waters. It also confirmed that the filament formed on the light side of the TMF, as isopycnal characteristics and stratification shifted markedly during the front crossing under the combined influence of southeasterly winds and the tidal ebb wave at 5 pm. Drifters independently captured the filament through trajectory diffraction and associated divergence-convergence patterns at its boundaries. ASV CTD transects during ebb revealed the filament as an even sharper horizontal density gradient within the uppermost meter, although the right filament border could not be fully resolved due to the transect length. ADCP profiles complemented these results by recording a full convergence signal across the filament, extending approximately 0.8 km horizontally and 4-8 m vertically. The repeated detection of the filament during both ebb and high tide underscores that it was a genuine TMF feature rather than a tidal current artifact.

Unlike dense filaments, which tend to form recurrently and persist over longer periods (McWilliams, 2009; Chrysagi et al., 2021; Garcia-Jove et al., 2022), light filaments appear far more volatile. They are scarcer, ephemeral, and strongly sensitive to freshwater input and tidal redistribution, which continuously reshape near-surface density gradients. Their buoyant positioning at the very top of the water column likely explains this fragility, since atmospheric forcing and tidal stirring act directly on the gradients that sustain them. Importantly, the gradients here were salinity-driven, in clear contrast to the temperature-driven dense filaments that dominate open-ocean settings (Mahadevan & Tandon, 2006; McWilliams et al., 2015; Jakes et al., 2023). This salinity control represents a unique pathway for filamentogenesis, potentially characteristic of shelf and estuarine systems such as the North Sea.

b. ASC patterns in the overlying first meter and lower NSL

The ASV CTD and ADCP measurements together provide a coherent view of filament ASC patterns, revealing three distinct phases that unfolded over the short span of only one hour. This rapid timescale highlights the transient nature of light filaments and the inherent difficulty of capturing their dynamics in the field. However, high-resolution multi-sensor and platform measurements help observing these processes as shown in this study.

In Phase I, the filament exhibited a sharp horizontal density gradient and a cone-like structure as Jakes et al. (2023) schematized. Yet the associated ASC in the NSL > 1 m, however, showed rather a filamentolysis pattern with downwelling occurring within the core and uplift at the flanks. Upwelling was induced by tidal modulation (see section c.) and along-front vertical velocity shear advecting density gradients differentially (Johnson et al., 2020). The vertical shear might be a consequence of down-front wind in the direction of the surface frontal jet, slowing down velocities in the upper and enhancing mixing in the lower NSL (Mahadevan and Tandon, 2006; Thomas et al., 2008). This instability acted to erode the filament, setting the stage for subsequent transitions in phases II and III.





Figure 6. Schematic process of filamentolysis-induced filamentogensis in the NSL. ASC associated with upper-NSL filamentogenesis (restratification) is colored pink, ASC associated with lower-NSL filamentolysis (mixing) is yellow.

In Phase II (Fig. 6), downwelling in the lower NSL weakened due to the decrease in density gradient strength, while signals of upwelling and divergence intensified. Early restratification established above 0.6 m with flattening of isopycnals on both boundaries of the filament core, indicating downwelling. The resulting ASC depicts the filamentogenesis of a light filament in the upcoming 40 min. The loss of buoyancy in the lower layer, therefore, fueled buoyancy production in the upper NSL, resulting in a two-layer coupled filamentolysis-filamentogenesis process.

In Phase III, these signatures intensified further: divergence broadened, upwelling strengthened, so that below 1 m the filament completely vanished from the water column. In the upper 0.6 m, restratification is fully established.

The complete filamentogenesis ASC pattern, characterized by upwelling in the core and downwelling at the flanks, was only captured in the upper 1 m (drifters, ASV CTD). Below 1 m, ADCP observations alone were insufficient to resolve the full






pattern, recording only the lower NSL filamentolysis. This highlights the importance of simultaneous, co-located ASV CTD and ADCP measurements for resolving the vertical structure of ASC in the NSL and for disentangling the coupling of submesoscale processes across layers.

#### c. Tidal modulation of filament formation and vertical velocities

The observed ASC patterns were strongly modulated by tides, which both enhanced and suppressed filamentary circulation depending on phase. Drifter data reveal that acceleration of the tidal flow induced divergence and upwelling, whereas deceleration toward low water produced convergence and downwelling. Due to the temporal offset between ASV and drifter deployments, the time series of drifter data was too short to resolve the coupling of tides with the filament at ebb tide. However, together with the data from the sensor chain, the tidal modulation of the filament could be resolved during high tide.

Destructive interference of tidal upwelling suppressed downwelling, reducing the filament's tendency to filamentolysis. Conversely, constructive interference occurred when tidal upwelling coincided with filamentary upwelling, amplifying upward velocities. This coincides with the reduction of the filament in the sensor chain signal that resembled the evolution of the filament in the upper NSL, supposing a similar process as in section b. This feedback further explains the asymmetric responses of the filament's borders: the left flank remained comparatively stable, while the right flank eroded in the vertical at a rate of 25 m·min<sup>-1</sup> and twice as fast as the left flank in the horizontal at a rate of 0.2 m·min<sup>-1</sup> under the constructive interference of filament and tidal downwelling after the current of the high tide started to decelerate.

Quantitatively, divergence values at the observed TMF reached 1-4 times higher than those reported in the Alboran Sea (Rypina et al., 2021; Garcia-Jove, 2022; Esposito et al., 2023) or the Bay of Bengal (Essink et al., 2022). While these exaggerated magnitudes may partly reflect overestimation by the area-based drifter method due to rapid tidal deformation of drifter triplets, they nonetheless point to strong tide-filament interactions. The resulting vertical velocities reached up to 50 m·d<sup>-1</sup> during divergence phases and down to -40 m·d<sup>-1</sup> during convergence phases. Comparisons between methods further highlight the tidal imprint, so that drifter-based estimates were roughly twice as large as ADCP-derived velocities, but converged to similar magnitudes once tidal displacement at the surface is removed from the drifter calculations.

# d. Implications for surface exchange and heat uptake

Once filamentogenesis was established in the upper meter, the system became more efficient at absorbing heat, with uptake doubling despite only weak net temperature changes. This enhanced capacity highlights the importance of light filaments as effective, short-lived hotspots of exchange at the ocean-atmosphere interface. Tidal modulation, salinity redistribution and rapid phase transitions may make them mediators of vertical exchange. While dense filaments dominate in the open ocean, light filaments may represent a likely important, yet less frequent, pathway for near-surface fluxes in shelf and estuarine seas.

https://doi.org/10.5194/egusphere-2025-4953 Preprint. Discussion started: 17 October 2025

© Author(s) 2025. CC BY 4.0 License.

## **5 Conclusion**


The study provides synoptic observation of a salinity-driven light filament on the submesoscale [O(0.1–10 m)] embedded within a tidal mixing front. By combining complementary datasets from sensor chains, drifters, ASV CTD transects, and ADCP profiles, a coherent sequence of filament evolution and associated ageostrophic secondary circulation (ASC) was identified that unfolded on timescales of approximately one hour. The three observed phases highlight the coexistence of filamentolysis in the deeper near-surface layer and filamentogenesis confined to the upper meter, revealing a tightly coupled, layer-dependent response that could only be captured through multi-instrument, high-resolution observations.

Unlike dense, temperature-driven filaments previously described in the open ocean, the light filament observed here was volatile, short-lived, and strongly modulated by tidal forcing. Its dynamics were further shaped by the redistribution of freshwater-derived salinity gradients, underscoring a unique estuarine pathway to filamentogenesis. The interplay of tidal currents with filamentary ASC produced constructive and destructive interference, explaining both the asymmetric erosion of filament flanks and the transition from strong lateral buoyancy gradients to restratification.

Restratification during filamentogenesis substantially enhanced the upper layer's capacity to store heat, linking submesoscale circulation directly to surface energy budgets. In this way, the dynamic processes like tidal modulation, atmospheric forcing, and salinity control that make light filaments scarce and prone to fast decay also make them important for vertical exchange and air—sea coupling. More broadly, resolving the short-lived but impactful dynamics of light filaments is essential for improving climate models. Their ability to mediate rapid exchanges of heat, gases, and momentum at the air—sea interface suggests that submesoscale processes may play a significant role in shaping upper-ocean stratification and regional climate feedbacks.

Taken together, the results expand the current paradigm of filament dynamics by demonstrating a salinity-driven mechanism for coupled light filamentogenesis and —lysis in shelf seas. Future work should quantify the frequency of such features in estuarine and coastal systems and assess their role in larger-scale balances of heat, gas exchange, and energy. To advance this, improved synoptic measurement strategies are needed, featuring congruent CTD and ADCP deployments that allow filamentary structure and ASC to be described more quantitatively, for instance, through frontal tendency calculations.

## **Appendices**

## 500 Appendix A


Since horizontal divergence is used to diagnose vertical motion, the vertical velocity in the water column can be determined. Restructuring the continuity equation  $\frac{\partial u}{\partial x} + \frac{\partial v}{\partial y} + \frac{\partial w}{\partial z} = 0$  results in


$$\begin{split} \Delta w &= - \int_{-z}^{z^0} \left(\frac{\mathrm{d}u}{\mathrm{d}x} + \frac{\mathrm{d}v}{\mathrm{d}y}\right) \mathrm{d}z \\ \leftrightarrow w(z_i) - w(z_{i-1}) &= \int_{z_i}^{z_{i-1}} \frac{\mathrm{d}u}{\mathrm{d}x} \mathrm{d}z \\ &= \left[\left(\frac{\mathrm{d}u}{\mathrm{d}x} + \frac{\mathrm{d}v}{\mathrm{d}y}\right)z\right]_{z_i}^{z_{i-1}} \\ &= \left(\frac{\mathrm{d}u}{\mathrm{d}x} + \frac{\mathrm{d}v}{\mathrm{d}y}\right)z_{i-1} - \left(\frac{\mathrm{d}u}{\mathrm{d}x} + \frac{\mathrm{d}v}{\mathrm{d}y}\right)z_i \\ &= \left(\frac{\mathrm{d}u}{\mathrm{d}x} + \frac{\mathrm{d}v}{\mathrm{d}y}\right)(z_{i-1} - z_i) \\ w(z_i) &= w(z_{i-1}) + \left(\frac{\mathrm{d}u}{\mathrm{d}x} + \frac{\mathrm{d}v}{\mathrm{d}y}\right)\Delta z, \end{split}$$


where  $w(z_i)$  is the vertical velocity in the target depth  $z_i$ ,  $w(z_{i-1})$  is the vertical velocity in the previous depth  $z_{i-1}$  and  $\Delta z$  is the difference between neighboring depths. For  $w(z_{i-1})$  of the drifters, the SSH (Fig. A1) is obtained from model data (CMEMS, 2024) to emulate the vertical velocity that arrives from the tidal wave

$$w(z_{i-1}) = \frac{1}{n} \sum_{k=1}^{n} ssh_k \cdot \frac{1}{\Lambda_t}, \tag{A2}$$

where  $ssh_k$  is the tidal amplitude and  $\Delta t=1$  h is the temporal resolution of the model data. Thus,  $w(z_{i-1})$  will be applied to (5) as hourly mean with a standard deviation of  $\pm 0.3$  m resulting in an error  $\Delta w(z_{i-1}) \approx 7 \text{ m} \cdot day^{-1}$ .


Figure A1. Hourly mean SSH obtained from model data (CMEMS, 2024) for the drifter deployments. The observation time of the filament is shaded in grey.

# Appendix B

Figure B1. Correlation of SSS from model data (CMEMS, 2024) and Ferrybox salinity regarding (a) salinity standard deviation, (b) time (UTC), (c) latitude and (d) longitude. The linear regression model results in a correlation of  $R^2 \approx 0.71$ .

# Appendix C


Figure C1. (top) Absolute wind speed  $v_{w,abs}$  and (bottom) wind direction at 10 m  $\theta_{w,abs}$  during the survey obtained from the weather station on RV Heincke. The grey shaded area highlights the period of the filament measurements.

## Appendix D

Figure D1. Vertical profiles of density from ship CTD during the observation period. Sections *t1-3* from the filament analysis of the ASV data (sections 3.3 and 3.4) are marked as vertical dashed lines.

## **Author contribution**

MA took the lead in writing the manuscript, conducting field measurements, processing and analysing data, and designing the figures. JM, LG and OW conducted field observations and data recording. OW and LJ processed the ASC CTD data. JM took the lead in supervising the work on the manuscript by streamlining the main concept and developing datasets and figures together with MA. THB supervised the project. All authors contributed to revising the manuscript by commenting on the manuscript and discussing the main concept and results of the study.

## 555 Competing Interests

The authors state that they have no conflict of interest.

https://doi.org/10.5194/egusphere-2025-4953 Preprint. Discussion started: 17 October 2025

© Author(s) 2025. CC BY 4.0 License.

EGUsphere Preprint repository

Acknowledgments

The authors would like to thank the captain and crew of RV Heincke and all the participants onboard for their support in

collecting this dataset.

**Financial Support** 





Financial support was received by the project "Sailing Intelligent Micro Drifter Swarms (Saimidris)" (grant no. VWZN3685)

and "The North Sea from space: Using explainable artificial intelligence to improve satellite observations of climate change

(North-SatX)" (grant no. VWZN3680). The study was conducted using resources of the "Biogeochemical processes and

Air–sea exchange in the Sea-Surface microlayer (BASS)" framework (DFG; grant no. 451574234).

**Data/Code Availability Statement** 

Remotely sensed sea surface salinity, derived from model data, are available from the Copernicus Marine Environment

Monitoring Service (CMEMS) at the following link https://doi.org/10.48670/moi-00054. CTD profile data

(https://doi.org/10.1594/PANGAEA.963643), ASV-mounted ADCP data (https://doi.org/10.1594/PANGAEA.973117) and

ASV CTD data (https://doi.pangaea.de/10.1594/PANGAEA.972989; in review) are published at Pangaea. Data from the

sensor chain and the drifters are still in preparation and will be published on PANGAEA as soon as post-processing is

completed. Raw data is available on request. Matlab-Code for computation and visualization of data is available on request.

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
