# Peer review of "Filamentogenesis and Filamentolysis of a Light Filament: Dynamic Processes in the Near-Surface Ocean Under Tidal Forcing"

_EGUsphere, 2025_

## Referee Comment (RC1)

**General Assessment**

The study investigates the presence and evolution of low-density filaments in the southern North Sea.

This manuscript presents a rich and valuable observational dataset combining an autonomous surface vehicle, surface drifters, and a drifting sensor chain to investigate the evolution of a low-density filament in the southern North Sea. The observations is a clear strength of the study and provides a rare opportunity to examine near-surface frontal processes under strong tidal forcing.

That said, the manuscript would benefit from a thorough revision of the text. At present, the presentation of the instrumentation and physical interpretation is often convoluted, and the main scientific narrative is obscured by a report-style structure. The central hypothesis regarding the formation and evolution of a low-density filament is not fully developed, and the physical mechanisms responsible for filament formation and decay are not clearly justified throughout the manuscript.

I believe the paper has strong potential. A clearer framing of the scientific questions, a more focused storyline, and a substantial reorganization of the text would already address many of the current weaknesses.

**Conceptual framing and terminology**

- The term *"light filament"* is used throughout the manuscript. While I understand the intention, the term may be misleading to readers outside the immediate submesoscale community, as "light" can imply luminosity rather than buoyancy. I suggest considering *"low-density filament"* or *"buoyant filament"*. This is a stylistic suggestion.

- Statements describing light filaments as *rare* and *dynamically significant* require explicit references and justification. It is not clear how rarity is defined here, nor relative to which class of filaments.

- The hypothesis of light-filament formation is introduced early, but the physical mechanism remains vague. It would help to clearly state what process is being tested, for example strain-driven frontogenesis, tidal modulation, or freshwater-induced buoyancy contrasts. Is not clear what is doing what.

- Several claims about filament lifetime, asymmetry, and dynamical importance appear without the supporting analysis. These arguments need evidence or should be clearly stated as .

**Introduction and background**

- Please add references to support the relationship between anticyclonic strain and the formation of buoyant filaments.

- When discussing filament lifetime, explicitly state why light filaments are hypothesized to be *short-lived*.

- The variability of the filament is mentioned, but not clearly defined. Specify whether this refers to changes in gradient sharpness, spatial position, depth extent, or temporal persistence.

- The description of filamentogenesis and filamentolysis is conceptually dense and not helpful in understanding how they occur in this scenario. A schematic or a direct reference to an existing schematic figure would substantially improve clarity.

- The term *"parcel deformation"* should be clearly defined or replaced with more specific kinematic language.

**Excessive use of acronyms**

The manuscript introduces many acronyms that are not strictly necessary and significantly hinder readability. Examples include TMF, NSL, DKL, and others. In several cases, spelling out the full term would improve clarity. I recommend being more selective when defining new acronyms.

**Data and methods**

- The motivation for using the CMEMS model for frontal detection should be clarified. Were satellite SST or SSS fields examined as an alternative or complement?

- Density variability appears to be dominated by salinity rather than temperature, yet the early sections give the opposite impression. This should be clarified consistently throughout the manuscript.

- Please clarify what is meant by *uniform resampling* of drifter trajectories.

- The description of the Delaunay triangulation procedure needs more detail, particularly why it is reinitialized at the start of each interval.

- The RMSD introduced in the context of SSH comparison is unclear. RMSD relative to what reference? Please clarify.

- The comparison between in situ measurements and CMEMS predictions raises questions about error propagation. A brief discussion of uncertainty introduced by this comparison would strengthen the analysis.

**Results and interpretation**

- Line ~282: clarify how large the remaining bias is after quality control. Here you say it could be significant. How can we trust this analysis then?

- Sharp discontinuities in Figure 3 panels b and c after ~12 hours need explanation.

- The formation of the filament is still not fully convincing. The processes are not clearly separated and justified.

- The role of tidal-induced vertical velocities is underdeveloped. There are no clear tidal diagnostics shown to support this interpretation.

- The manuscript repeatedly refers to freshwater input as a key driver, but the freshwater source and its spatial structure are never clearly described.

- Statements regarding heat transport and heat uptake are speculative unless directly supported by quantitative analysis. If such analysis is not included, these claims should be toned down (Line ~465 and other places).

- The conclusion that light filaments are more ephemeral than dense filaments is not sufficiently demonstrated. There is no comparison of lifetimes or statistics that would support a general claim.

**Minor Comments and Editorial Issues**

- Line ~24: please indicate chapter and page number for the cited book reference.

- Line ~114: typographical correction for "~2 km".

- Line ~214: sentence structure is unclear and should be revised.

- When introducing rotated coordinates, explicitly state that x′ refers to the rotated reference frame.

- Line ~270: Avoid starting paragraphs with phrases such as *"Figure X shows"* (https://brushingupscience.com/2019/11/04/dont-start-paragraphs-with-figure-n-shows/).

**Figures**

- Figure captions should be self-contained. Several figures rely heavily on the main text to explain what is shown.

- All units should be written using square brackets, for example [kg m$^{-3}$], rather than separated by slashes.

- In Figure 5, the density panels would benefit from adding a grid to improve readability.

**Final Recommendation**

This manuscript presents a unique and valuable dataset and addresses an interesting physical problem. However, in its current form, the scientific narrative is difficult to follow, and several interpretations are insufficiently supported. I strongly encourage a major revision focused on clarifying the hypothesis, simplifying the structure, reducing acronyms, and strengthening the physical interpretation. With these changes, the paper could make a meaningful contribution to the study of near-surface frontal dynamics in tidally dominated seas.